# Adherence and Persistence to Biological Drugs for Psoriasis: Systematic Review with Meta-Analysis

**DOI:** 10.3390/jcm11061506

**Published:** 2022-03-09

**Authors:** Eugenia Piragine, Davide Petri, Alma Martelli, Agata Janowska, Valentina Dini, Marco Romanelli, Vincenzo Calderone, Ersilia Lucenteforte

**Affiliations:** 1Department of Pharmacy, University of Pisa, 56126 Pisa, Italy; eugenia.piragine@farm.unipi.it (E.P.); alma.martelli@unipi.it (A.M.); vincenzo.calderone@unipi.it (V.C.); 2School of Specialization in Hospital Pharmacy, University of Pisa, 56126 Pisa, Italy; 3Department of Clinical and Experimental Medicine, University of Pisa, 56126 Pisa, Italy; davide.petri@unipi.it; 4Department of Dermatology, University of Pisa, 56126 Pisa, Italy; agata.janowska@unipi.it (A.J.); valentina.dini@unipi.it (V.D.); marco.romanelli@unipi.it (M.R.)

**Keywords:** psoriasis, biological drugs, anti-TNF-α, anti-IL-17, anti-IL-12/23, adherence, persistence

## Abstract

Despite the large number of biologics currently available for moderate-to-severe psoriasis, poor adherence and persistence to therapy represent the main issues for both the clinical and economic management of psoriasis. However, the data about adherence and persistence to biologics in psoriasis patients are conflicting. Our aim was to produce summary estimates of adherence and persistence to biologics in adult patients with psoriasis. We performed a systematic review and meta-analysis of observational studies, searching two databases (PubMed and Embase). Sixty-two records met the inclusion criteria, and a meta-analysis was conducted on fifty-five studies. Overall, the proportion of adherent and persistent patients to biological therapy was 0.61 (95% confidence interval: 0.48–0.73) and 0.63 (0.57–0.68), respectively. The highest proportions were found for ustekinumab, while the lowest ones were found for etanercept. The proportions of adherence and persistence to biological drugs in psoriasis patients are sub-optimal. Notably, both proportions largely differ between drugs, suggesting that a more rational use of biologics might ensure better management of psoriasis.

## 1. Introduction

Psoriasis affects about 30 million adults worldwide [1]. Genetic factors, as well as lifestyle (smoking, alcohol consumption, and diet), certain drugs, environmental factors, and various metabolic conditions, can promote the development and progression of psoriasis [2,3]. Although the etiopathogenesis of psoriasis is multifactorial, its clinical manifestation mainly results from both uncontrolled keratinocyte proliferation and the overproduction of inflammatory mediators, such as tumor necrosis factor-α (TNF-α), interleukin (IL)-17, IL-12, and IL-23. In particular, the activation of these pro-inflammatory molecules triggers a vicious circle that progressively exacerbates psoriasis [2].

Due to its peculiar clinical manifestation, psoriasis has a negative psychological impact on patients, deeply affecting their quality of life [4]. Moreover, patients with psoriasis usually have several comorbidities that further aggravate their clinical condition [1]. Therefore, adequate pharmacological treatment might ameliorate both disease severity and, indirectly, the psychosocial sphere of the individual.

The therapeutic armamentarium currently available for the management of psoriasis is mainly represented by anti-inflammatory drugs and immunomodulators. In particular, topical (i.e., corticosteroids, vitamin D3 derivatives, and keratolytic products) and systemic drugs, such as methotrexate and retinoids, are commonly used in the mild-to-moderate forms of psoriasis, while targeted biological drugs are recommended for patients with severe forms who fail to respond to first-line therapy. TNF-α inhibitors were the first biologics to obtain marketing authorization and reimbursement for psoriasis, and they include etanercept (ETN), infliximab (INF), and adalimumab (ADA) [5]. Other biologics are IL17A inhibitors (ixekizumab, IXE; secukinumab, SECU) [6] and ustekinumab (UST), which is an anti-IL12/23 human monoclonal antibody [7].

Despite the large number of therapeutic options for the clinical management of psoriasis, two key contributors to both treatment failure and scarce relapse control are poor adherence and persistence to therapy. Adherence reflects “the extent to which a patient acts in accordance with the prescribed interval, and dose of a dosing regimen”, while persistence, also known as drug survival, is “the duration of time from initiation to discontinuation of therapy” [8]. In addition, suboptimal adherence and persistence deeply impact the economic management of psoriasis in healthcare systems [9], especially for the more expensive drugs (i.e., biologics). Therefore, improving medication-taking behaviors may help patients to better control therapy, as well as limiting the economic health expenditure. Currently, the data about adherence and persistence to biological therapy in psoriasis patients are scarce and conflicting, and previous systematic reviews, although quite recent [8,10], do not provide an exhaustive and quantitative synthesis of the literature. Moreover, real-world data about adherence and persistence to individual biologics are discordant, thus hindering the rational use of these drugs in clinical practice.

Hence, the aim of this systematic review and meta-analysis is to provide overall, updated adherence and persistence proportions to biologics, as well as reporting a stratification of results based on the individual biological drugs.

## 2. Materials and Methods

The protocol for this systematic review and meta-analysis was registered in the PROSPERO database (CRD42021245065).

### 2.1. Eligibility Criteria

We included prospective, retrospective, and cross-sectional observational studies evaluating adherence and persistence (or drug survival) to biologic drugs among participants aged 18 years or older with psoriasis. We considered studies irrespective of patient gender, comorbidities, or concomitant drugs. Biological drugs belonging to the following 3 classes were considered: TNF-α inhibitors (ETN, ADA, INF); IL17A inhibitors (IXE, SECU); and IL12/23 inhibitors (UST). The outcomes were adherence and persistence to biologics, as reported in the included studies.

### 2.2. Information Sources and Search Strategy

We searched Medline and EMBASE for studies published from inception to 18 January 2021. The search strategy (Appendix A) reports psoriasis as the first term; drug or therapy adherence, persistence, compliance, and switching as the second term; and the considered biologic drugs as the third term (etanercept, ustekinumab, adalimumab, infliximab, ixekizumab, and secukinumab). The three terms were combined using the Boolean operator “AND”.

### 2.3. Selection Process

Titles and abstracts of papers identified by the search strategy were screened by two authors independently, E.P. and D.P. Each paper was categorized as not relevant or potentially included according to the eligibility criteria. Any disagreement was discussed with another author, E.L.

The full text of the potentially includible articles was retrieved or, if not available, directly requested from the authors of the study. Two authors (E.P. and D.P.) checked the full texts for the eligibility criteria and excluded studies not fitting them.

The selection process was managed using bibliographic management software Mendeley Desktop (v1.19.6, Mendeley Ltd., London, UK).

### 2.4. Data Extraction Process

We extracted the following information: study design, outcome (adherence or persistence), and objective; number and general characteristics of participants included in the studies, such as age, gender, comorbidities, and concomitant drugs (drugs used for the treatment of psoriasis, as well as other drugs); definition of adherence/persistence as reported in the study; number of adherent/persistent patients; and reasons for discontinuation/switching. Furthermore, the data relating to any stratifications were retrieved. The data extraction was carried out by two authors independently, E.P. and D.P., and any discrepancies were resolved through consultation with a third reviewer, E.L.

For the data collection, spreadsheet software Microsoft Excel was used (version 2102 build 13801.20864).

### 2.5. Study Risk of Bias Assessment

The methodological quality of included studies was assessed according to risk of bias in prevalence studies developed by Hoy et al. [11]. The tool considers ten domains concerning characteristics of prevalence studies, each rated in terms of risk of bias and applicability to research question. Risk of bias was judged from 0 (high risk) to 10 (low risk). The risk of bias was evaluated by two authors independently, E.P. and D.P., and any discrepancies were resolved through consultation with a third reviewer, E.L.

### 2.6. Effect Measures

We evaluated the study-specific prevalence of adherence or persistence (drug survival) to biologics by calculating the proportion of adherent or persistent subjects on the total number of participants for each study. Where the study provided adherence/persistence as a percentage or where the non-adherence/non-persistence was provided, appropriate calculations were performed.

### 2.7. Synthesis Methods

As adherence and persistence refer to two different concepts that cannot be matched and pooled, we separately analyzed these parameters, as previously reported by others [12]. In detail, three outcomes were evaluated in our meta-analysis: (1) adherence; (2) good adherence, generally reported as the medication possession ratio (MPR) or proportion of days covered (PDC) ≥ 80%; and (3) persistence.

Study-specific means of adherence were pooled using random effect models and the generic inverse variance method. Study-specific adherence/persistence proportions were pooled using random effect models with Freeman–Tukey transformation.

The heterogeneity for both methods was quantified through the Higgins heterogeneity index (I^2^) and was tested through the chi-square test for mean adherence and Cochran’s Q test for adherence/persistence proportion.

Subgroup analyses were conducted according to study design (retrospective observational, prospective observational, or cross-sectional), type of biologics, the type of biologic users (biological-naïve subjects, i.e., subjects who have never used a biological drug, and biological-experienced subjects, i.e., subjects who have already had experience with this type of treatment), and study quality (high quality, score ≥ 8, vs. low quality, score < 8). Differences between groups was considered statistically significant if the heterogeneity test was significant.

*p*-value < 0.10 was considered statistically significant.

The “metagen” and “metaprop” routines within the META package in R (version 4.12) was used for analyses [13].

## 3. Results

### 3.1. Systematic Review

A flowchart of the search is presented in Figure 1. We identified 1285 records from the PUBMED search and 2698 from EMBASE. In total, 62 studies, including 169,371 participants, met the inclusion criteria and were included in the qualitative synthesis. Three studies [14,15,16] did not show data on persistence or the number of persistent patients, while one did not show data on adherence [17]; two studies were conducted on patients not only affected by psoriasis [18,19] and did not report adherence data for psoriasis patients; one study [20] did not report the number of patients treated with each biological drug but only adherence as percentage. Fifty-five studies [21,22,23,24,25,26,27,28,29,30,31,32,33,34,35,36,37,38,39,40,41,42,43,44,45,46,47,48,49,50,51,52,53,54,55,56,57,58,59,60,61,62,63,64,65,66,67,68,69,70,71,72,73,74,75] on 161,748 participants were included in the quantitative synthesis (meta-analyses).

In 13 studies [24,25,28,32,34,35,38,39,55,64,66,70,75], the sample was composed of patients with other chronic inflammatory autoimmune conditions, including osteoarticular diseases (such as ankylosing spondylitis and rheumatoid arthritis), bowel diseases (such as ulcerative colitis and Crohn’s disease), and psoriatic arthritis. The extraction of data, in this case, focused on the cohorts of patients suffering from psoriasis regardless of other conditions.

Among the included studies (Table 1), 5 studies presented data on adherence [22,27,33,44,53], 16 studies on good adherence [21,25,26,27,32,33,35,40,44,46,49,53,63,68,71,73], 46 studies on persistence data [21,23,24,25,27,28,29,30,31,33,34,36,37,38,39,41,42,43,44,45,47,48,49,50,51,52,53,54,55,56,57,58,59,60,61,62,64,65,66,67,68,69,70,72,74,75], and 8 studies reported data on both adherence and persistence [21,25,27,33,44,49,53,68]. Regarding study design (Appendix A), 51 were retrospective cohort studies [15,18,19,21,22,23,24,25,26,27,28,29,30,31,32,33,34,35,36,37,38,39,40,41,42,43,44,45,46,47,48,49,51,52,53,54,55,56,57,59,60,61,62,64,66,67,68,70,72,74,75], 5 were prospective cohort studies [14,50,58,65,69], and 6 were cross-sectional studies [16,17,20,63,71,73]. The mean age of the participants was 47 years, of which about 45% were female. Thirty-two studies reported no use of concomitant drugs [14,16,17,18,19,21,23,24,25,26,28,29,30,31,34,37,39,42,51,52,55,58,61,62,64,67,68,70,71,73,75]. Twenty-four studies presented data on biological-naïve patients [26,27,28,29,30,31,32,37,38,39,41,43,46,48,50,51,53,55,60,61,66,68,74,75], while four studies [37,39,50,75] reported data on biological-experienced patients. Twenty-eight studies reported data on ADA adherence/persistence [26,27,29,31,32,33,35,36,39,40,48,49,50,51,52,53,55,61,63,64,65,66,67,68,69,70,74,75], fifteen on INF [27,29,36,39,47,49,50,52,53,55,57,64,67,70,75], twenty-five on ETN [22,26,27,29,30,31,32,36,39,40,48,49,50,51,52,53,55,56,63,66,67,68,70,74,75], four on IXE [28,33,44,54], ten on SECU [24,28,32,34,44,59,61,68,69,70], and twenty-one on UST [23,26,27,31,32,39,40,48,50,51,53,60,61,63,64,67,68,69,70,72,75]. Finally, forty-five studies were included in biological drug subgroup analysis [21,23,24,26,27,28,29,30,31,32,33,34,35,36,39,40,42,43,44,46,47,48,49,50,51,52,53,54,55,56,59,60,61,63,64,65,66,67,68,69,70,72,73,74,75] and twenty-eight in experienced/naïve subgroup analysis [26,27,28,29,30,31,32,35,37,38,39,41,43,46,48,50,51,53,55,60,61,66,68,74,75].

Table 1 shows details on adherence, good adherence, and persistence. The measures were highly heterogeneous: 5 studies gave the mean of adherence using MPR or PDC measures defined during different periods; 16 studies gave the proportion of adherent patients by mainly using (11 out of 16) the cut-off of 80% of MPR or PDC measures defined during different periods; 46 gave the proportion of persistent patients by mainly using (38 out of 64) discontinuation or switch or no gap (from 7 to 150 days) concepts defined during different periods.

### 3.2. Risk of Bias in Studies

Seventeen studies [18,25,26,27,32,39,45,46,48,49,53,59,61,62,67,68,69] obtained a total score of 10 in quality assessment based on the scale of Hoy et al. [11], while three studies [34,42,73] scored less or equal than 6 points. Details on single domains can be found in Appendix A.

### 3.3. Results of Synthesis

#### 3.3.1. Adherence

The meta-analysis conducted on five studies including 11,129 patients showed a mean adherence of 65% (95% confidence interval, CI: 61–70%, Appendix A) with considerable heterogeneity (I^2^ = 99%). Among 16 studies including 45,252 patients, the proportion of good adherence was 61% (48–73, Figure 2), with considerable heterogeneity (I^2^ = 100.0%). Only 2 out of 16 studies reported the reasons for non-adherence, which were loss of efficacy and adverse events. Qualitative descriptions of the reasons are shown in Appendix A.

Regarding the stratification according to the type of biologic drug, the highest adherence proportion (Table 2, Figure 3 and Figure 4) was observed for UST (72%, 48–91), followed by INF (63%, 44–80), ADA (62%, 47–76), SECU (52%, 35–68), ETN (50%, 36–65), and, finally, IXE (46%, 43–48). The difference between groups was statistically significant (*p*-value = 0.04). ADA, ETN, and UST represent the three biological drugs most considered in the included studies, as the use of each of them was evaluated in a considerable number of studies compared to the others: 10 studies for the first drug [26,27,32,33,35,40,49,53,63,68], 8 for the second [26,27,32,40,49,53,63,68], and 7 for the third [26,27,32,40,53,63,68].

There were differences stratifying by study design, with the cross-sectional design (85%; 55–100) showing a higher adherence compared to the retrospective cohort design (54%; 43–66) (*p*-value from subgroup test = 0.06) (Table 2 and Appendix A); however, only 3 studies had a cross-sectional design in contrast with 13 retrospective cohort studies. There were no differences stratifying by biological-naïve patients and not-specified patients (*p*-value = 0.24) (Table 2 and Appendix A) or stratifying by risk of bias (*p*-value = 0.40) (Appendix A).

#### 3.3.2. Persistence

The meta-analysis conducted on 46 studies including 156,801 patients showed a persistence proportion of 63% (57–68, Figure 5), with considerable heterogeneity (I^2^ = 100%). Less than half of the studies (19 out of 46) reported the reasons for drug discontinuation or switching. The most common reasons were loss of efficacy and adverse events (nine studies) followed by ineffectiveness (three studies). Qualitative descriptions of the reasons are shown in Appendix A.

Regarding the stratification according to the type of biological drug, the highest persistence (Table 3, Figure 6 and Figure 7) was found for UST (77%, 70–84), followed by SECU (72%, 58–84), IXE (70%, 52–85), INF (64%, 60–68), ADA (57%, 50–63), and ETN (53%, 42–65). The heterogeneity between groups was statistically significant (*p*-value < 0.01). ADA, ETN, and UST represent the three biological drugs most considered in the included studies, as the use of each of them was evaluated in a considerable number of studies compared to the others: 22 studies for the first drug [29,31,36,39,48,50,51,52,55,61,64,65,66,67,69,70,74,75], 19 for the second [29,30,31,36,39,48,50,51,52,55,56,66,67,70,74,75], and 17 for the third [23,31,39,48,50,51,60,61,64,67,69,70,72,75].

Different persistence proportions (*p*-value from heterogeneity test = 0.05) were observed among 21 studies on biological-naïve patients (56%, 49–64), 4 studies on biological-experienced patients (50%, 35–65), and 25 studies where it was not specified (67%, 60–74) (Table 3 and Appendix A). There were also statistical differences in study design stratification (*p*-value from subgroup test = 0.07) (Table 3 and Appendix A). However, only 4 studies had a cross-sectional design in contrast with 42 retrospective cohort studies. There were no statistical differences in the risk of bias stratification (*p*-value = 0.78) (Appendix A).

## 4. Discussion

We systematically reviewed data from 55 studies including 161,748 psoriatic patients and showed that 61% of patients were adherent to biologic therapy and 63% were persistent. Our findings are consistent with those reported in previous studies. In a systematic review on inflammatory bowel disease [76], 23–62% of patients were found adherent to biologics. Another systematic review on rheumatoid arthritis [77] reported a median adherence value of 63% for both ETN and ADA. Finally, two recent meta-analyses on psoriasis showed that 66% of patients were persistent at 1 year [78] and 53.2% at two years [79].

In the studies included in our systematic review, the main reported reasons for drug discontinuation, switching, or non-adherence were loss of efficacy and adverse events. However, many other aspects could affect the patient’s behavior toward biological therapy. The female gender, recent disease onset, smoking, the presence of comorbidities, and a lack of efficacy of the previous treatments have been reported as predictors of non-persistence/non-adherence [42,71,80,81,82]. On the contrary, the presence of psoriatic arthritis has been generally associated with sustained drug survival of biological agents [69,81].

The variability in the included studies is reflected in the heterogeneity of our analysis. We found that biological-naïve patients were more persistent than biological-experienced patients. However, only four studies evaluated persistence in biological-experienced patients. Moreover, we observed a high percentage of adherent and persistent patients among cross-sectional and prospective cohort studies, respectively, compared to retrospective cohort ones. This is expected, even if only three studies evaluating adherence had a cross-sectional design and four studies evaluating persistence had a prospective cohort design. We did not investigate whether the inclusion of different definitions of the concepts of adherence and persistence influenced our results because few studies used the same definition. This represents a limitation of our study, as well as other meta-analyses aimed at pooling adherence and persistence. The proposal of a unified set of definitions might be useful to make the results of future studies more consistent and comparable [83].

At present, the data about adherence and persistence to individual biological drugs are quite scarce. This evaluation is essential to guide clinicians toward a more rational therapeutic choice, which is fundamental for both medical and economic purposes. In our study, the highest adherence was found for the human antibody UST (72%), followed by INF (63%), ADA (62%), SECU (52%), ETN (50%), and IXE (46%). Similar proportions were found for persistence as, in descending order, they were UST with 77%, SECU with 72%, IXE with 70%, INF with 64%, ADA with 57%, and ETN with 53%.

The variability in both adherence and persistence to specific biologics could derive, to a minimal extent, from the differences in the efficacy of treatments, which can reasonably affect patients’ satisfaction and, consequently, adherence/persistence to therapy. Future studies are required to elucidate on comparative efficacy because few data derived from direct “head-to-head” comparisons, and short-term efficacy outcomes were mainly evaluated [84,85,86,87]. A role for body mass index (BMI) in the patient’s attitude toward biological treatment has been recently proposed [88,89]. Indeed, the efficacy of TNF-α inhibitors and UST is reduced in obese/overweight patients with psoriasis [90,91,92], with consequences for both adherence and persistence [93,94].

The difference between adherence and persistence to biological therapies can be certainly explained by discussing the origin, therapeutic class, administration route and timing, and toxicity profile of the biologics. Firstly, the immunogenic potential of chimeric antibodies (i.e., INF) might cause acute anaphylactic reactions following infusion, as well as hypersensitivity reactions (such as influenza-like syndrome, local skin reactions, and pyrexia) [95]. These phenomena can be counteracted with concomitant immunosuppressive therapy, with serious consequences on patient compliance and medication adherence/persistence [96]. On the contrary, pharmacological treatment with fully human antibodies (ADA, SECU, and UST) is less associated with anti-drug antibody production, although residual immunogenicity has been reported even for the most innovative biological drugs [97].

All biological drugs share the common risk of mild-to-moderate adverse events, including headache, cutaneous and upper respiratory tract infections, and injection site reactions, which can dramatically reduce quality of life [98]. Generally, these reactions do not require additional therapy, but they can be prevented by combining biological drugs with immunomodulators [99], with predictable detrimental effects on the patient’s compliance. Notably, TNF-α inhibitors are generally associated with a higher risk of severe infections, and they can induce hypersensitivity reactions [100] and dermatological disorders [101]. Hence, the peculiar toxicity profile of TNF-α inhibitors might explain, at least in part, the sub-optimal medication adherence and persistence to ETN, ADA, and INF in psoriatic patients.

Among the TNF-α inhibitors, ETN is self-administered using pre-filled syringes or pens up to twice a week. Both self-administration and short intervals between administrations might reduce compliance [102]. ADA is administered subcutaneously every 2 weeks. Therefore, the interval between administrations is longer than that reported for ETN, partially justifying the better adherence and persistence proportions to ADA rather than ETN. Finally, INF is intravenously administered at weeks 0, 2, and 6 after initiation and then at an 8- to 12-week interval [103]. The outpatient administration of INF ensures periodic support is provided to psoriasis patients, as well as contributing to a more assiduous monitoring of therapy by clinicians. Importantly, patients with scheduled appointments do not forget to take drugs, and they do not make mistakes, which instead can occur in self-administered therapy.

Concerning IL17A inhibitors, SECU is self-administered once a week for 4 weeks and then every 4 weeks [104], while IXE is self-administrated every 2 weeks for the first 12 weeks and then once a month [105].

The highest adherence and persistence proportions were found for the IL12/23 inhibitor UST (72% for adherence and 77% for persistence). UST is characterized by high efficacy in the treatment of moderate-to-severe forms of psoriasis [106] and a favorable safety profile [107]. Moreover, it is administered every 12 weeks, exclusively under the guidance of an experienced physician. Both the longest administration interval and the supervision of a healthcare provider might favorably impact adherence and persistence [108]. The subcutaneous administration of UST using pre-filled syringes or pens might also explain the wide difference in the adherence and persistence proportions from another biological drug administered in hospitals or clinics, namely, INF (63% vs. 64%, respectively). Indeed, the latter therapy requires a slow 2 h infusion followed by an additional monitoring period of 2 h; it is a demotivating protocol that might partially contribute to scarce medication adherence and persistence.

In accordance with our findings, a comparative meta-analysis showed that UST has the longest persistence at 5 years after initiation compared with TNF-α inhibitors (ETN, ADA, and INF) [79]. In a meta-analysis of real-world evidence, UST appeared as the biological drug less frequently discontinued due to loss of efficacy [78], thus confirming its clinical relevance in the pharmacological treatment of psoriasis. On the contrary, ETN showed the worst persistence and the highest number of therapy interruptions for low efficacy, supporting the results of our meta-analysis.

Even if UST is one of the most expensive biological drugs, it is endowed with one of the most favorable cost-efficacy profiles among the biological drugs for psoriasis [109]. Indeed, both sustained adherence and persistence and a sporadic dose regimen reduce the direct costs of treatment in the long term [110], but great attention should be paid to obese patients requiring high dosage [111]. In addition, UST is associated with minor indirect costs for the healthcare system, as it reduces hospital visits for non-responders; treatment failure; and resultant drug switching, which is associated with a 7–17% increase in annual costs [112]. There are, however, some crucial aspects that must be considered before initiating biological therapy with UST. Of course, UST must be avoided in patients with hypersensitivity to this biological drug or any of the excipients [110]. Moreover, health insurance coverage does not apply in all cases in real clinical practice. This latter aspect is reported to be responsible for short-term intermittent treatment with UST [113], as uninsured patients cannot afford the economic burden of continuous treatment with this biologic drug. Hence, the expansion of insurance coverage might ameliorate both patients’ satisfaction and adherence/persistence toward biological therapy. Finally, patient preferences should also be considered before starting therapy with UST, as involving patients in treatment decisions can influence both adherence to treatment and the outcomes of therapy [114].

## 5. Conclusions

The adherence and persistence to biological therapy in psoriasis patients are sub-optimal; however, the initial therapeutic choice might be crucial to ensure better medication adherence/persistence. Psoriasis patients are more adherent and persistent to therapies with a favorable safety profile and that are characterized by less frequent administrations (i.e., UST). However, several aspects regarding comorbidities, insurance coverage, patient preferences, and costs must be considered before initiating therapy with UST. We suggest that constant real-life therapeutic discussions between health providers (dermatologists, general practitioners, pharmacists, and nurses) and their patients, as well as specific support programs, might promote the optimal levels of adherence and persistence to biological drugs for both clinical and economic purposes.

Our study has several strengths, including the high number of studies identified and the large sample size, which gives consistency to the results. Our study also has some limitations, such as having considered work from all over the world; therefore, it cannot be excluded that adherence and persistence to treatment may have a link with reimbursement policies that vary from country to country. In addition, the follow-up period was variable from study to study, although most papers were aligned in considering 12 months as the follow-up period. We are also aware that we had to exclude some studies because of the lack of usable data, even though they met all inclusion and exclusion criteria. The age and sex of the participants could influence adherence/persistence; however, in the studies included in our meta-analysis, patients were very similar in terms of age and sex. Finally, although we included all drugs approved before May 2021 (data of our literature search) to manage moderate-to-severe psoriasis, we did not include studies investigating new groups of drugs, for example, selective inhibitors of IL-23 in a recently published study [115], and this represents a limitation of our systematic review.

## Figures and Tables

**Figure 1 jcm-11-01506-f001:**
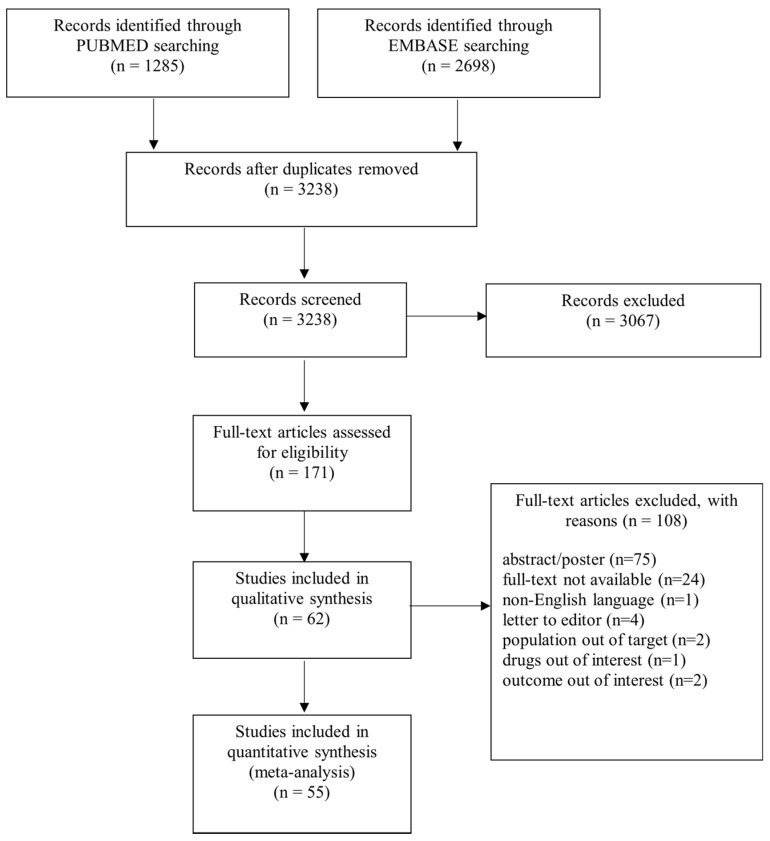
Flowchart of search.

**Figure 2 jcm-11-01506-f002:**
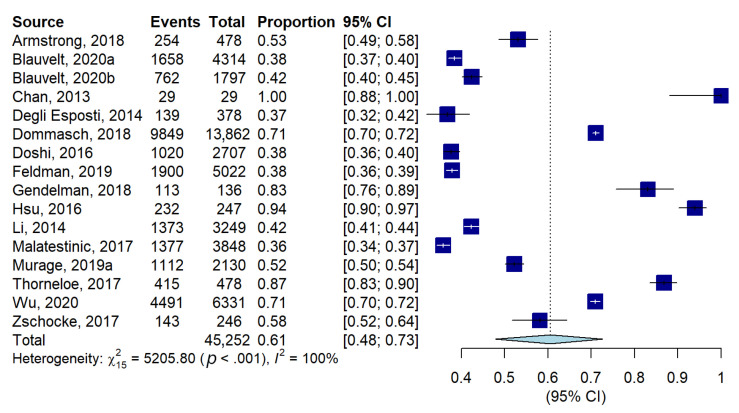
Forest plot of proportions, and their 95% confidence intervals, of adherent patients.

**Figure 3 jcm-11-01506-f003:**
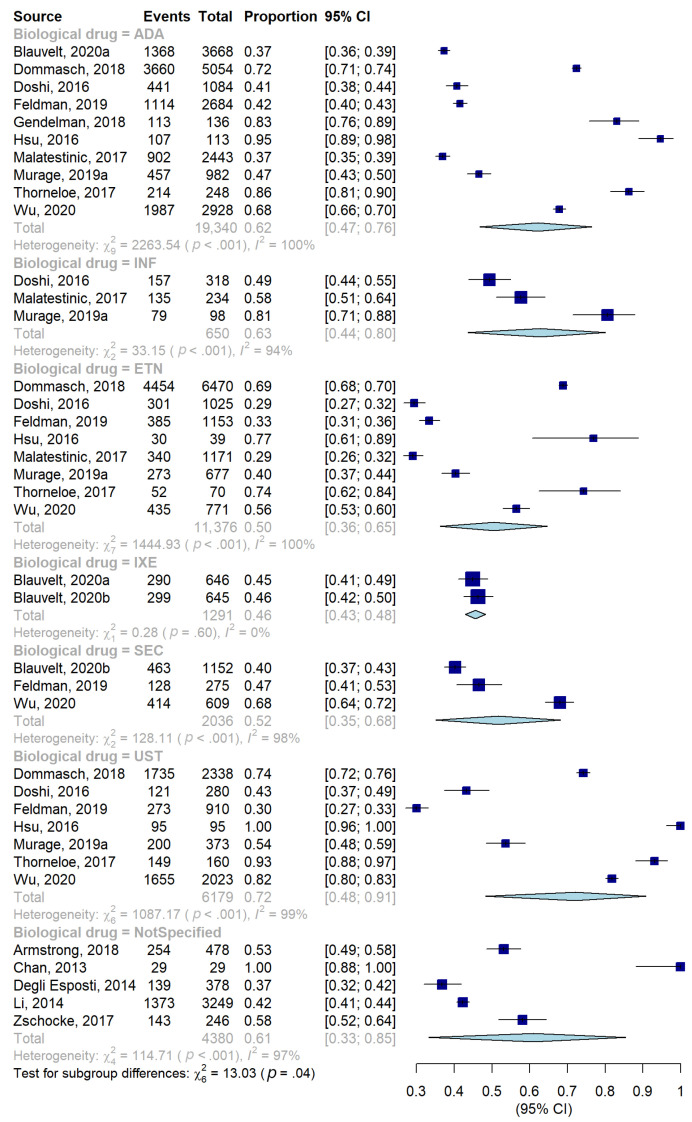
Forest plot of proportions, and their 95% confidence intervals, of adherent patients stratified according to biological drugs.

**Figure 4 jcm-11-01506-f004:**
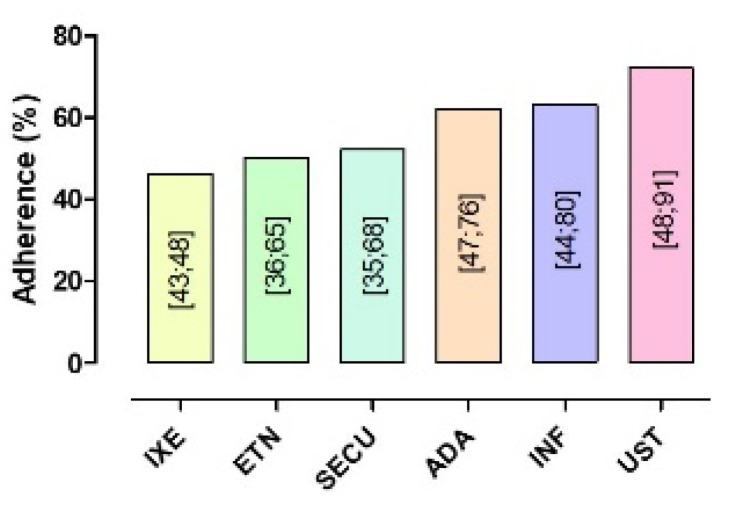
Percentage of adherent patients according to biological drugs. Confidence intervals (95%) are reported within the vertical bars.

**Figure 5 jcm-11-01506-f005:**
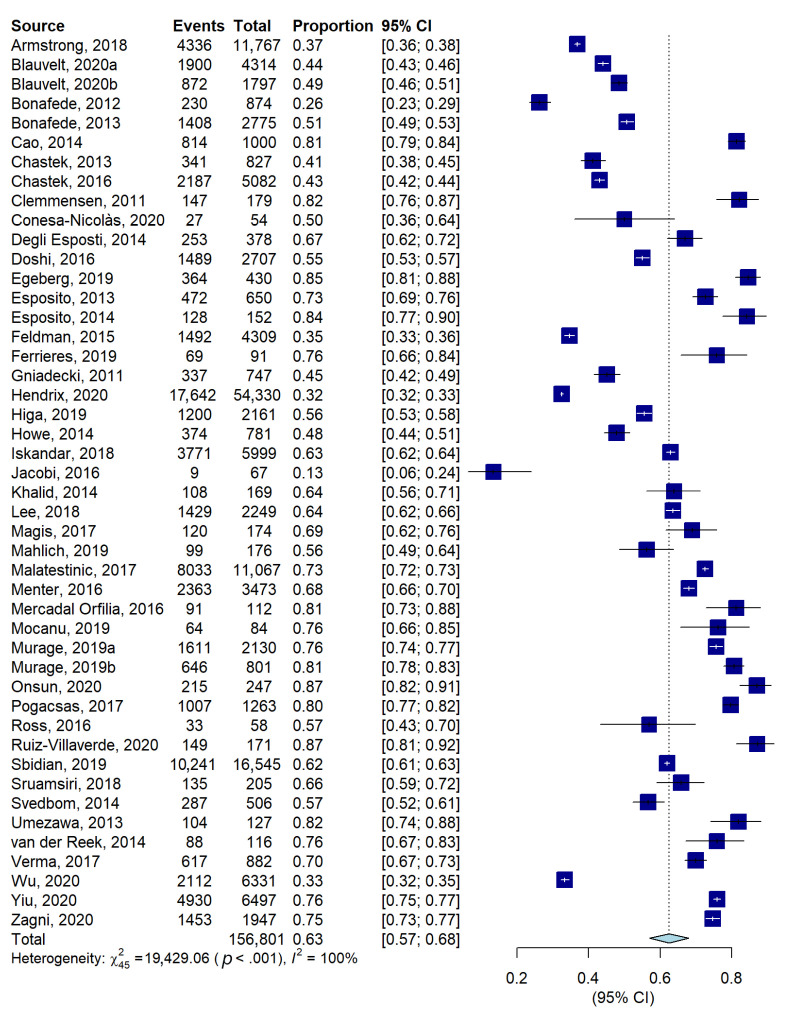
Forest plot of proportions, and their 95% confidence intervals, of persistent patients.

**Figure 6 jcm-11-01506-f006:**
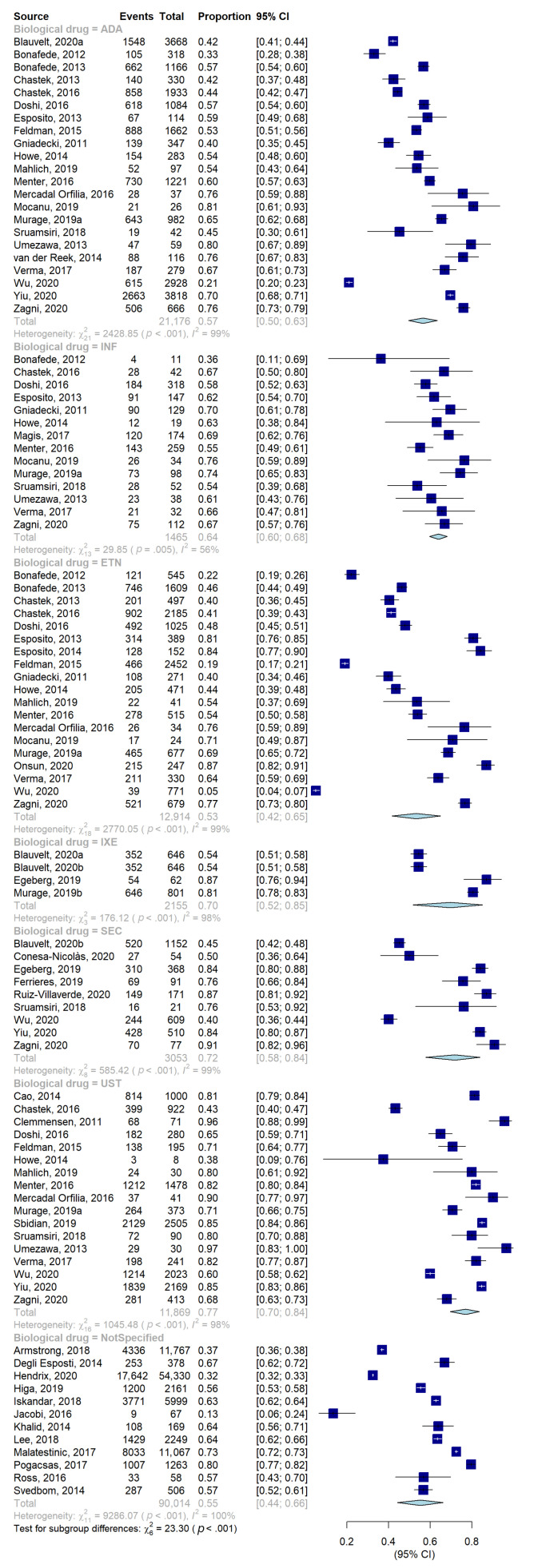
Forest plot of proportions, and their 95% confidence intervals, of persistent patients stratified according to biological drugs.

**Figure 7 jcm-11-01506-f007:**
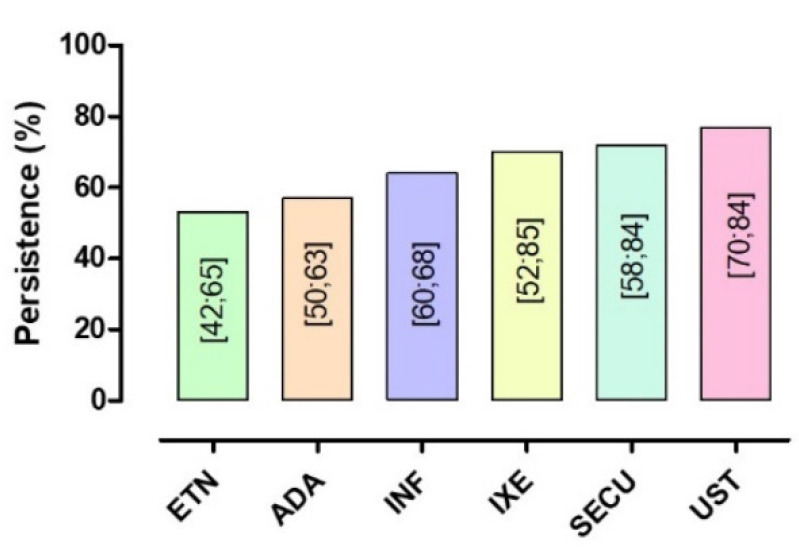
Percentage of persistent patients according to biological drugs. Confidence intervals (95%) are reported within vertical bars.

**Table 1 jcm-11-01506-t001:** Details of calculation methods in considered outcomes.

	No. of Studies (No. of Patients)
Adherence	
MPR/PDC mean	
during a period of 12 months	2 (4832) [27,53]
during a period of >12 months	3 (6297) [22,33,44]
Good adherence	
MPR/PDC ≥ 80%	
during a period of 12 months	6 (29,256) [25,26,27,49,53,68]
during a period of >12 months	5 (11,516) [32,33,35,40,44]
Other definitions during different or not-specified periods ^a^	5 (4480) [21,46,63,71,73]
Persistence	
No discontinuation or gap ^a^ or switch	
during a period of <12 months	2 (1179) [23,72]
during a period of 12 months	24 (114,864) [24,27,28,31,37,38,39,43,45,48,49,51,53,54,55,60,61,62,64,65,66,68,74,75]
during a period of >12 months	11 (24,246) [29,33,34,41,42,44,50,56,58,59,67,69]
during a not-specified period	1 (84) [52]
Still on treatment	
after a period of <12 months	1 (378) [25]
after a period of 12 months or more	4 (2336) [30,36,47,57]
Other definitions during different or not-specified periods	2 (13,714) [21,70]

^a^ different permissible gaps (from 7 to 150 days).

**Table 2 jcm-11-01506-t002:** Pooled proportions of adherent patients stratified according to study design, type of biological drug, and type of patient.

	No. of Studies	No. of Patients	Adherence, %, [CI 95%]	I^2^	Q	*p*-Value for Heterogeneity within Strata	*p*-Value for Heterogeneity between Strata
Overall	16	45,252	61 [48; 73]	99.7%	5205.80	0	
Study design							
Cross-sectional	3	753	85 [55; 100]	98%	89.03	<0.0001	0.06
Retrospective cohort	13	44,499	54 [43; 66]	100%	4905.70	<0.0001	
Biological drug							
ADA	10	19,340	62 [47; 76]	100.0%	2263.54	0	0.04
ETN	8	11,376	50 [36; 65]	100.0%	1444.93	<0.0001	
INF	3	650	63 [44; 80]	94.0%	33.15	<0.0001	
UST	7	6179	72 [48; 91]	99.0%	1087.17	<0.0001	
IXE	2	1291	46 [43; 48]	0.0%	0.28	0.5976	
SECU	3	2036	52 [35; 68]	98.0%	128.11	<0.0001	
Not specified	5	4380	61 [33; 85]	97.0%	129.88	<0.0001	
Type of patient							
Biological naïve	6	33,301	52 [39; 65]	99.8%	3107.32	0	0.29
Not specified	12	12,912	63 [47; 78]	99.1%	1198.03	<0.0001	

**Table 3 jcm-11-01506-t003:** Pooled proportions of persistent patients stratified according to study design, type of biological drug, and type of patient.

	No. of Studies	No. of Patients	Persistence, %, [CI 95%]	I^2^	Q	*p*-Value for Heterogeneity within Strata	*p*-Value for Heterogeneity between Strata
Overall	46	156,801	63 [57; 68]	100.0%	19,429.06	0	
Study design							
Retrospective cohort	42	146,657	62 [56; 68]	100.0%	16,496.95	0	0.07
Prospective cohort	4	10,144	71 [63; 77]	96.1%	76.69	<0.001	
Biological drug							
ADA	22	21,176	57 [50; 63]	99.0%	2428.85	0	<0.001
ETN	19	12,914	53 [42; 65]	99.0%	2770.05	0	
INF	14	1465	64 [60; 68]	56.0%	29.85	0.0049	
UST	17	11,869	77 [70; 84]	98.0%	1045.48	<0.001	
IXE	4	2155	70 [52; 85]	98.0%	176.12	<0.001	
SECU	9	3053	72 [58; 84]	99.0%	585.42	<0.001	
Not specified	12	90,014	55 [44; 66]	100.0%	9286.07	0	
Type of patient							
Biological naïve	21	66,821	56 [49; 64]	100.0%	5408.54	0	0.05
Biological experienced	4	43,097	50 [35; 65]	100.0%	1638.78	<0.001	
Not specified	25	46,583	67 [60; 74]	100.0%	5961.35	0	

## Data Availability

The data that support the findings of this study are available from the corresponding author upon reasonable request.

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
