# Peer review of "Adherence and Persistence to Biological Drugs for Psoriasis: Systematic Review with Meta-Analysis"

_jcm, 2022, doi:10.3390/jcm11061506_

Round 1

Reviewer 1 Report

This article is a systematic review of the adherence and persistence of biologic agents (etanercept , infliximab, adalimumab, ustekinumab, secukinumab and ixekizumab) for psoriasis.   It is recommended that the authors state logically and concisely what they infer from the results.  

It is redundant to give the same overview of psoriasis in the abstract, introduction, and discussion, so the authors should keep it short in the appropriate sections. For example, in the introduction section, the description of biologics and Table 1 seem excessive.

  Since methods of measuring adherence and persistence may vary in each article, they should be described in detail in the results section. For example, the authors included articles on drug survival as references; as the drug survival rate decreases over time, it is necessary to clearly state how the percentage of persistence was calculated besides mentioning it in the conclusion section.   On page 2, line 4, it is histopathologically incorrect to describe the erythema of psoriasis as 'the sclerotic'.

Author Response

REVIEWER #1

This article is a systematic review of the adherence and persistence of biologic agents (etanercept , infliximab, adalimumab, ustekinumab, secukinumab and ixekizumab) for psoriasis. It is recommended that the authors state logically and concisely what they infer from the results.  

Reply: We modified conclusion as follow (lines 475-486)

Our meta-analysis showed that psoriasis patients are more adherent and persistent to therapies with a favorable safety profile, and characterized by less frequent administra-tions (i.e., IL12/23 inhibitors). UST is also one of the most cost-effective biological drugs, as the infrequent administrations lead to cost savings in the long period for the healthcare systems. However, several aspects regarding comorbidities, insurance coverage, patient preferences and costs must be considered before initiating therapy with UST. This me-ta-analysis aims to represent a potential guide for treatment decision-making in this field, suggesting that constant real-life therapeutic discussions between health providers and their patients, as well as specific support programs, might promote optimal levels of adherence and persistence to biological drugs for both clinical and economic purposes.

It is redundant to give the same overview of psoriasis in the abstract, introduction, and discussion, so the authors should keep it short in the appropriate sections. For example, in the introduction section, the description of biologics and Table 1 seem excessive.

Reply: We revised paper and delete redundant information in the Abstract (deleting 1st sentence with definition of psoriasis), in the Introduction (deleting Table 1 with the description of biologics), and in the Discussion (deleting mechanism of action of drugs and re-organizing text in according with all Reviewers).

Since methods of measuring adherence and persistence may vary in each article, they should be described in detail in the results section. For example, the authors included articles on drug survival as references; as the drug survival rate decreases over time, it is necessary to clearly state how the percentage of persistence was calculated besides mentioning it in the conclusion section.  

Reply: We added a table (new Table 1) and the following paragraph in results section with details on adherence and persistence measures (lines 195-201):

“Table 1 shows details on adherence, good adherence and persistence. Measures were highly heterogeneous: five studies gave the mean of adherence using MPR or PDC measures defined during different periods; 16 studies gave the proportion of adherent patients using mainly (11 out of 16) the cut-off of 80% of MPR or PDC measures defined during different periods; 46 gave proportion of persistent patients using mainly (38 out of 64) discontinuation or switch or no gap (from 7 to 150 days) concepts defined during different periods

On page 2, line 4, it is histopathologically incorrect to describe the erythema of psoriasis as 'the sclerotic'.

Reply: We apologize for the mistake and changed the sentence as follow (lines 45-46):

“Therefore, an adequate pharmacological treatment might ameliorate both the disease severity and, indirectly, the psychosocial sphere of the individual.”

Reviewer 2 Report

The authors have done a good job on summarizing the current literature on drug adherence and persistence in psoriasis patients. There are currently several meta-analyses addressing the same topic, but the current paper adds valuable insights for it includes a large number of published articles (56 studies), and includes data for secukinumab (which was not the case for the study from Lin et al (Sci Rep 8, 16068 (2018)). One drawback of this article is that the reasons for drug discontinuation or switching were not assessed. Nonetheless,

  1. The authors used the word “persistence” throughout this manuscript. Although the use of “persistence” seems appropriate, some readers are more familiar with the term “drug survival”. Thus, I suggest that the authors define “persistence” in the early sections and state that it also refers to “drug survival”.
  2. Based on the findings of this study, one could argue that choosing ustekinumab over other biologics as the first biological drug would be highly recommended. The authors could describe some circumstances where other biological drugs could be preferred over ustekinumab (comorbidities, patient preferences, cost, insurance coverage, etc.).

  1. Adherence to therapy is also largely influenced by the severity of psoriasis as well as the overall duration of treatment before the initiation of biological therapy. The authors need to mention other factors that affect adherence.

  1. Some minor mistakes and typos need to be corrected:

Line 261: ‘2’ in I2 should be in upper index.

Line 329: “scarce and conflicting”

Line 367: “the contribute”, do the authors mean “contribution?”

Lines 390, 494: “noteworthy” could be replaced by “of note” or “notably”, depending on the authors’ intention.

Line 399: “also causes a paradoxical”

Line 408: “Crohn’s disease”

Line 493: “low efficacy”

Line 509: “periodic access to hospitals and clinics”, I believe the authors mean “less frequent visits to hospitals and clinics”. This sentence could be corrected for clarity.

Line 519: “for both clinical and economic purposes”

Author Response

REVIEWER #2

The authors have done a good job on summarizing the current literature on drug adherence and persistence in psoriasis patients. There are currently several meta-analyses addressing the same topic, but the current paper adds valuable insights for it includes a large number of published articles (56 studies), and includes data for secukinumab (which was not the case for the study from Lin et al (Sci Rep 8, 16068 (2018)).

Reply: We thank the Reviewer for appreciating our work.

One drawback of this article is that the reasons for drug discontinuation or switching were not assessed.

Reply: We added a table in the Appendix (new Table A3) with qualitative description on reasons for drug discontinuation or switching and added the following sentences in the Results:

“Only 2 out of 16 studies reported the reasons for non-adherence, which were loss of effi-cacy and adverse events. Qualitative description of reasons is shown in Table A3.” (lines 215-217)

“Less than half studies (19 out of 46) reported the reasons for drug discontinuation or switching. The most common were loss of efficacy and adverse events (9 studies) fol-lowed by ineffectiveness (3 studies). Qualitative description of reasons is shown in Table A3.” (lines 251-253)

And the following in the Discussion:

“In the studies analyzed in our systematic review, the main reasons for drug discon-tinuation, switching or non-adherence were loss of efficacy and adverse events.”. (lines 302-303)

Nonetheless,

  1. The authors used the word “persistence” throughout this manuscript. Although the use of “persistence” seems appropriate, some readers are more familiar with the term “drug survival”. Thus, I suggest that the authors define “persistence” in the early sections and state that it also refers to “drug survival”.

Reply: We modified the following sentence in the Introduction:

[…] persistence, also known as drug-survival, reflects “the duration of time from initiation to discontinuation of therapy(line 64)

and the following in the Methods:

“We included prospective, retrospective, and cross-sectional observational studies evaluating adherence and persistence (or drug-survival) to biologic drugs […]” (line 82)

“We evaluated the study-specific prevalence of adherence or persistence (drug-survival) to biologics by calculating […]” (line 123)

  1. Based on the findings of this study, one could argue that choosing ustekinumab over other biologics as the first biological drug would be highly recommended. The authors could describe some circumstances where other biological drugs could be preferred over ustekinumab (comorbidities, patient preferences, cost, insurance coverage, etc.).

Reply: We added the following paragraph describing some circumstances whether UST should not be the best choice.

In fact, even if UST is one of the most expensive biological drugs, both sustained adherence and persistence and sporadic dose regimen reduce the direct costs of treatment in the long-term period [113]. In addition, UST is associated with minor indirect costs for the health-care system. Indeed, the efficacy of UST reduces hospital visits for non-responders, treatment failure and resultant drug switching, which is associated with a 7-17% increase in annual costs [114] and reduced quality of life [31]. Therefore, even if UST is associated with higher initial costs compared to other biologics, there are im-portant long-term cost savings. However, there are some crucial aspects that must be considered before initiating biological therapy with UST. Of course, UST must be avoided in patients with hypersensitivity to this biological drug or any of the excipients [113]. Moreover, as obesity is a common comorbidity in psoriasis patients [115], is fun-damental to account the influence of body weight on the cost-effective profile of UST therapy. Indeed, treating patients over 100 kg generally requires a 90 mg of UST per dose to achieve results comparable to 45 mg of UST. As costs associated to a chronic therapy with UST 90 mg are approximately $5,327-$6,993 higher, UST is not a cost-effective therapeutic option in overweight/obese patients [116]. Moreover, health insurance cov-erage does not apply in all cases in real clinical practice. This latter aspect is reported to be responsible for short-term intermittent treatment with UST [117], as uninsured pa-tients cannot afford the economic burden of continuous treatment with this biologic drug. Hence, the expansion of insurance coverage might ameliorate both patients’ satis-faction and adherence/persistence toward biological therapy. Finally, patient preferences should be also considered before starting therapy with UST. A recent web-based survey conducted in Japanese patients with rheumatoid arthritis showed that they prefer self-injected (66.8% of patients) to in-hospital injected biologics (18.0%), as they are asso-ciated with low frequency hospital visits and flexible administrations [118]. Hence, it is important to reconsider the role of patients in the initial choice of biologic drug, as in-volving patients in treatment decisions can influence both adherence to treatment and outcomes of therapy [119].” (lines 435-461)

  1. Adherence to therapy is also largely influenced by the severity of psoriasis as well as the overall duration of treatment before the initiation of biological therapy. The authors need to mention other factors that affect adherence.

Reply: We added the following paragraph mentioning female gender, recent diseases on-set, psychosocial factors, comorbidity and lack of efficacy of previous treatment as possible risk factors for non-adherence/non-persistence. 

“However, many other aspects could affect the patient’s behavior towards biological therapy, including sex, severity of disease, and psychosocial factors. For instance, female gender, recent disease onset, and being a current smoker have been reported as predic-tors of non-persistence [82,83]. On the contrary, presence of psoriatic arthritis has been generally associated to sustained drug survival of biological agents [71,83]. The major predictors of non-adherence to biologics in psoriasis patients, instead, are the presence of comorbidities and a lack of efficacy of the previous treatments [44,73].many other aspects could affect the patient’s behavior towards biological therapy, in-cluding sex, severity of disease, and psychosocial factors. For instance, female gender, recent disease onset, and being a current smoker have been reported as predictors of non-persistence [82,83]. On the contrary, presence of psoriatic arthritis has been generally associated to sustained drug survival of biological agents [71,83]. The major predictors of non-adherence to biologics in psoriasis patients, instead, are the presence of comorbidi-ties and a lack of efficacy of the previous treatments [44,73].” (lines 303-310)

  1. Some minor mistakes and typos need to be corrected:

Line 261: ‘2’ in I2 should be in upper index.

Reply: done. (line 250)

Line 329: “scarce and conflicting”

Reply: done. (line 288)

Line 367: “the contribute”, do the authors mean “contribution?”

Reply: Yes, thank you. We replaced term as suggested. (line 346)

Lines 390, 494: “noteworthy” could be replaced by “of note” or “notably”, depending on the authors’ intention.

Reply: done. (lines 433)

Line 399: “also causes a paradoxical”

Reply: the sentence with the typo has been deleted.

Line 408: “Crohn’s disease”

Reply: the sentence with the typo has been deleted.

Line 493: “low efficacy”

Reply: the sentence with the typo has been deleted.

Line 509: “periodic access to hospitals and clinics”, I believe the authors mean “less frequent visits to hospitals and clinics”. This sentence could be corrected for clarity.

Reply: No, more frequent visits. We replaced sentence accordingly. (lines 475-476)

Line 519: “for both clinical and economic purposes”

Reply: done. (lines 485-486)

Reviewer 3 Report

The authors decided to conduct a systematic review along with a meta-analysis of adherence and persistence of biological drugs used in psoriasis.

The topic is interesting with a potential clinical application, but manuscript requires some improvements:

  • Please provide in Table 1 the correct, modern names of biological drug groups, i.e. TNF inhibitors, IL-17A inhibitors.
  • The definition of adherence and persistance should be discussed in more detail in the introduction or the methodology
  • The discussion is too long, it should be more focused on discussing the results obtained, among other things:

(A) the influence of different definitions of the concepts of adherence and persistence in publications on the obtained results

(B) compare the results with previous work on adherence and persistance

(C) comparison with real world data results from registers

(D) explanation why the analysis lacks new groups of drugs - selective inhibitors of IL-23 (anti-p19)

(E) whether patients' BMI could have a potential impact on adherence and persistance

Author Response

REVIEWER #3

The authors decided to conduct a systematic review along with a meta-analysis of adherence and persistence of biological drugs used in psoriasis. The topic is interesting with a potential clinical application, but manuscript requires some improvements:

Please provide in Table 1 the correct, modern names of biological drug groups, i.e. TNF inhibitors, IL-17A inhibitors.

Reply: We deleted Table 1 according to suggestion of Reviewer #1, however, we used modern name of biologics through manuscript as follow:

TNF-α inhibitors instead of anti TNF-α agents

IL-17A inhibitors instead of anti-IL17A agents

IL12/23 inhibitors instead of anti-IL12/23 human monoclonal antibody

The definition of adherence and persistence should be discussed in more detail in the introduction or the methodology

Reply: We added a table in the Appendix (new Table A3) with qualitative description on reasons for drug discontinuation or switching and added the following sentences in the Results:

“Only 2 out of 16 studies reported the reasons for non-adherence, which were loss of effi-cacy and adverse events. Qualitative description of reasons is shown in Table A3.” (lines 215-217)

“Less than half studies (19 out of 46) reported the reasons for drug discontinuation or switching. The most common were loss of efficacy and adverse events (9 studies) fol-lowed by ineffectiveness (3 studies). Qualitative description of reasons is shown in Table A3.” (lines 251-253)

And the following in the Discussion:

“In the studies analyzed in our systematic review, the main reasons for drug discon-tinuation, switching or non-adherence were loss of efficacy and adverse events.”. (lines 302-303)

The discussion is too long, it should be more focused on discussing the results obtained, among other things:

(A) the influence of different definitions of the concepts of adherence and persistence in publications on the obtained results

(B) compare the results with previous work on adherence and persistance

(C) comparison with real world data results from registers

(D) explanation why the analysis lacks new groups of drugs - selective inhibitors of IL-23 (anti-p19)

(E) whether patients' BMI could have a potential impact on adherence and persistence

Reply: We shorted, as suggested also by Reviewer #1, and re-organized the Discussion. In particular, we added paragraphs with:

  • limitation due to the inclusion of studies with different definitions of the concepts of adherence and persistence (A) (lines 320-323)
  • comparison of our results with those of other systematic reviews (B): we focused the comparisons of our results with those reported in other systematic review on adherence or persistence to biologic drugs (lines 293-301)
  • limitation due to the lack of new groups of drugs (D) (lines 499-502)
  • the role of BMI (E) (lines 373-378)

As concern request C, our meta-analysis included mainly real word data (52 out of 55), thus, we think comparison with real word studies is redundant. We limited comparison with results reported in other systematic reviews and meta-analyses, as requested in the point B.

Round 2

Reviewer 1 Report

I think the results section has improved. But the introduction and discussion sections are still too long.  The discussion section is mostly based on quotes from previous papers. The authors should state what can be derived from the results of the study. 

On page14, line 372, the authors state that "Indeed , TNF-α inhibitors [100–102] can induce body weight increase in patients with chronic plaque psoriasis, thus reducing adherence  and persistence towards TNF-α inhibitors [103]  " but I think that is an incorrect quote, as weight gain is not generally a problem with TNF inhibitors.

Author Response

REVIEWER #1

I think the results section has improved. But the introduction and discussion sections are still too long.  The discussion section is mostly based on quotes from previous papers. The authors should state what can be derived from the results of the study. 

Reply: We shorted Introduction (from 47 to 40 lines) and Discussion (from 184 to 123), as suggested.

From results we derive that “Adherence and persistence to biological therapy in psoriasis patients are sub-optimal, however initial therapeutic choice might be crucial to ensure better medication adherence/persistence. Psoriasis patients are more adherent and persistent to therapies with a favorable safety profile, and characterized by less frequent administrations (i.e., UST). However, several aspects regarding comorbidities, insurance coverage, patient preferences and costs must be considered before initiating therapy with UST. We suggest that constant real-life therapeutic discussions between health providers (dermatologists, general practitioners, pharmacists, and nurses) and their patients, as well as specific support programs, might promote optimal levels of adherence and persistence to biological drugs for both clinical and economic purposes”.

We rephrased the conclusion as above (lines 388-397)

On page14, line 372, the authors state that "Indeed , TNF-α inhibitors [100–102] can induce body weight increase in patients with chronic plaque psoriasis, thus reducing adherence  and persistence towards TNF-α inhibitors [103]  " but I think that is an incorrect quote, as weight gain is not generally a problem with TNF inhibitors.

Reply: We revised sentence as follow (lines 313-316):

“A role for body mass index (BMI) in the patient’s attitude towards biological treatment has been recently proposed. Indeed, the efficacy of TNF-α inhibitors and UST is reduced in obese/over-weight patients with psoriasis with consequence on both adherence and persistence

Reviewer 2 Report

The authors have addressed all my queries. The manuscript has been much improved. 

Author Response

REVIEWER #2

The authors have addressed all my queries. The manuscript has been much improved. 

Reply: Thank you.